# Understanding Large Language Models Through the Lens of Dataset Generation

## Abstract

There has been increased interest in using Large Language Models (LLMs) for text dataset generation subject to a desired attribute, e.g., for use in downstream fine-tuning or training. These works generally focus on a single quality metric of the generated text, typically accuracy on a downstream task. However, this fails to consider whether the model even has the ability to faithfully model the data distribution of the desired real-world domain. In contrast, in this work, we additionally focus on important distributional metrics agnostic to the downstream task, such as data diversity and faithfulness. We show that even in simple domains, generated datasets reveal inherent trade-offs between these metrics across models and training regimes. Further, we find that our metrics not only describe the generated dataset, but also capture key aspects of the underlying model. This allows us to characterize the generated datasets, individual models and by comparison the properties of different model families and training paradigms. By focusing on sub-distributions well-represented in the training data of LLMs, we can, for example, show that popular instruction-tuning techniques strongly decrease the LLM's text generation abilities, with respect to distributional aspects like diversity.

## 1 Introduction

In recent years, large language models (large LMs, LLMs), often called foundation models, have become the state-of-the-art on many NLP tasks and beyond. These models can achieve outstanding performance on many tasks, often without any adaption or only with minimal prompting (Brown et al., 2020; Rae et al., 2021; Chowdhery et al., 2022; Touvron et al., 2023a; Bommasani et al., 2021).

**Need for Task-Specific Data And Dataset Generation** The direct application of LLMs can be effective and has the advantage that users do not have to do any additional training or data collection beforehand. However, in practice, smaller custom models that were trained on task-specific data still outperform LLMs, both in terms of task accuracy and hardware efficiency (Ye et al., 2022a; Hsieh et al., 2023; Gao et al., 2023). Recent work also focuses on fine-tuning LLMs themselves on task-specific data, either via standard training (Hu et al., 2022; 2023; Chen et al., 2021), self-improvement (Bai et al., 2022b; Wang et al., 2022b; Haluptzok et al., 2023; Wang et al., 2022a), reinforcement learning from human feedback (RLHF) (Stiennon et al., 2020; Bai et al., 2022a; Ouyang et al., 2022) or even via in-context learning (Brown et al., 2020). Fundamentally, however, all of these methods again require the construction of task-specific datasets, which can be a cumbersome and expensive.

In response to this, recent works explore the use of LLMs themselves to automatically generate such datasets (Ye et al., 2022a; Gao et al., 2023; Ye et al., 2022b; Meng et al., 2022; Schick and Schütze, 2021; Josifoski et al., 2023; Chia et al., 2022; Bonifacio et al., 2022). Here, LLMs are prompted to generate synthetic data for a particular task, which can then be used to train, fine-tune or prompt a model, thereby avoiding the need for manual data collection.

Despite promising early results in LLM-based data generation, prior work does not fully explore important distributional characteristics of the resulting synthetic datasets in comparison to real-world data, or how data quality differs across different LLMs and sampling strategies. However, going forward, achieving a better understanding of these factors is very important as it (1) provides insights on the actual data modeling capabilities of different LLMs as they are deployed more widely, and (2) can help inform and improve synthetic dataset generation in general.

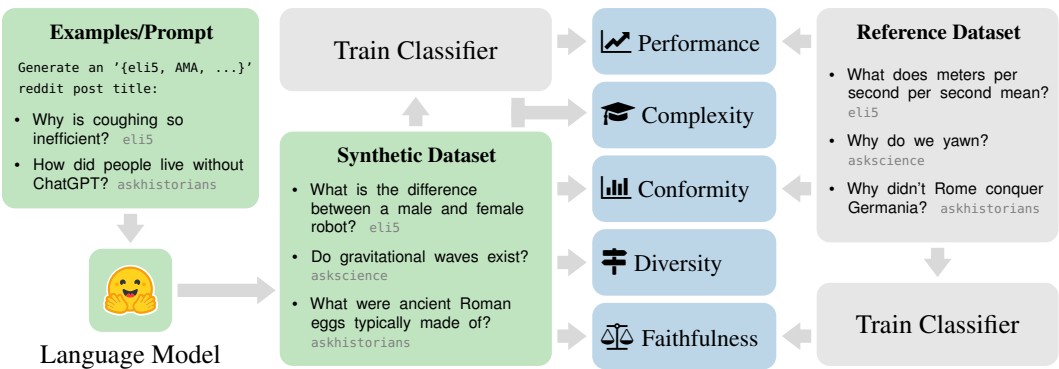

Figure 1: Overview of our data generation (shown in green, ●) and evaluation pipeline. Arrows show dependencies. We prompt language models (🤗) with examples or instructions to generate a synthetic dataset. We then compare the resulting data to real-world reference samples using several distributional metrics (shown in blue, ●), and thereby assess the model's generative capabilities.

**Metrics For Dataset Quality**   Providing an in-depth analysis of the quality of generated datasets requires analyzing data from various angles. To address this, we propose a multi-faceted evaluation framework, showcased in Fig. 1. We prompt an LLM with examples or instructions to generate a synthetic dataset, which we then compare to a real-world reference dataset, for a wide range of different metrics. Most prior work only uses performance on a downstream task, often classification, as the fundamental metric to characterize synthetically generated datasets. While task performance is important, it does not necessarily transfer to other tasks and does not allow for an effective comparison between models. To account for this, and inspired by Ye et al. (2022a;b); Gao et al. (2021b), our framework goes beyond just task performance and relies on four extra characteristics that encompass further aspects of dataset quality: As included in Fig. 1, we examine *complexity* (🎓), i.e. how complex or simple the synthetic dataset is based on classifier performance, *conformity* (📊), i.e., how well the synthetic dataset reflects the distribution of the (real) reference dataset, *diversity* (🎏), how distinct individual samples in the synthetic dataset are, and *faithfulness* (⚖️), i.e., how well the synthetic samples fits the desired data domain, in addition to standard task *performance* (📈).

**Understanding LLM Dataset Generation**   To better understand the generative abilities of LLMs, we apply our framework to four simple, but representative domains, each of which is chosen such that we can be sure that it is well-represented in the training data of common LLMs, and that a real-world reference dataset is readily available. This allows us to assess the overall data generation capabilities of LLMs with respect to these domains, and to compare different LLMs and sampling strategies. We evaluate the generative abilities of 22 LLMs in total – corresponding to different model families, fine-tuning methodologies, training datasets, available openly or via the OpenAI API – and a wide range of sampling configurations, including zero- and few-shot strategies.

**Inherent Trade-Offs**   In an in-depth analysis, we reveal underlying tradeoffs between distributional metrics, which we find to apply broadly across all domains and models. We observe a quadratic relationship between diversity and conformity, but that diversity and faithfulness are inversely correlated. Moreover, conformity and faithfulness exhibit a very high correlation, but our experiments also show that small variations in this regard very much characterize a model's generative behavior.

**Comparing Models**   We also compare across models and, e.g., find that LLAMA-2's generative abilities mainly improve over LLAMA-1 on conformity, while keeping other characteristics constant. With respect to training paradigm, we find that instruction-tuned models generally exhibit higher faithfulness, but much lower diversity, conformity and complexity when compared to their vanilla base model counterparts. Increasing sampling temperature with instruction-tuned models can bring them more on-par with vanilla models, but even then, neither paradigm clearly dominates a general performance ranking. Lastly, we repeatedly find that OpenAI's instruction-tuned models exhibit very different generative behavior when compared to open instruction-tuned models like LLAMA-2, thus hinting at notable differences with respect to their (proprietary) training data and procedure.

## 2 SYNTHETIC DATASET GENERATION

We first discuss the relevant background of language modeling and synthetic dataset generation, and the concrete data generation procedure we rely on.

**(Large) Language Models for Text Completion** In this work, we consider language models capable of performing text completion. While our focus lies on large language models, all we assume is a simple text generation interface. We thus use the term language model throughout the rest of this paper. Further, we consider models relying on different training regimes, including *vanilla LMs* trained on a standard text completion objective Brown et al. (2020); Touvron et al. (2023a); Almazrouei et al. (2023) and *instruction-tuned LMs*, trained via fine-tuning or reinforcement learning with human feedback Ouyang et al. (2022); Touvron et al. (2023b).

**Synthetic Dataset Generation with LMs** Due to their strong generative capabilities, recent work has started to incorporate LMs for automated dataset generation, either to directly train downstream models Taori et al. (2023); Chiang et al. (2023) or as part of a self-improvement process Haluptzok et al. (2023). Given a domain $\mathcal{D}$, the goal is to construct a dataset $S_{\mathcal{D}}$ of samples that fit domain $\mathcal{D}$. Interesting choices for $\mathcal{D}$ include text of sentiment, forms of speech, instruction following and examples of e.g. puzzle solving. If this generation process additionally leverages some (small) existing reference dataset $R_{\mathcal{D}}$, it can also be understood as a form of LM-based data augmentation.

In this work, we specifically consider *dataset generation for classification*. More specifically, we construct synthetic datasets $S_{\mathcal{D}}$, given a suitable instructive or few shot Brown et al. (2020) prompt. As $\mathcal{D}$, we choose common domains like movie reviews or posts in online forums, because we can safely assume that these lie in-distribution for all considered LMs and human-curated reference datasets $R_{\mathcal{D}}$ are readily available for comparison. More importantly, common data domains allow us to measure the extend to which the LMs have learned a good representation of these data domains during training. For each domain, e.g. movie reviews, we define a set of classes $\{c_1, \ldots, c_n\}$, e.g. positive, undecided, negative, etc. To generate synthetic data, we prompt an LM to produce new dataset samples that fit the different classes $c_i$, using class- and domain-specific prompts $p_i$.

**Generative Pipeline** We illustrate the data generation pipeline we consider, in the left part of Fig. 1 (in green ●). Here, we generate samples for the domain $\mathcal{D} = \texttt{posts of a subreddit}$ (type of online forum) for subreddits of different topics, e.g. `explainlikeimfive` (`eli5`), a community where explanations in child-appropriate language are shared and `askhistorians`, where historians answer questions. We consider both zero-shot and few-shot sampling. In the zero-shot setting we prompt a model with `"A question that appeared on the subreddit 'eli5'"`. Adapting this for each of the classes $\{c_1, \ldots, c_n\}$, allows us to obtain a wide variety of labeled samples fitting domain $\mathcal{D}$. In the few-shot setting we additionally provide samples from the reference dataset $R_{\mathcal{D}}$. We experiment with varying sampling temperatures (higher temperature leading to higher entropy samples) to further analyze the tradeoffs present within an LM.

Using this generative pipeline, we construct synthetic classification datasets for a number of exemplary domains (see §4), for which we also obtain (human-curated) real datasets as comparison point.

## 3 EVALUATION FRAMEWORK

We now introduce our evaluation framework for the correct representation of common data domains $\mathcal{D}$ in LLMs. For this purpose, we compare datasets against a valid representation of a data domain $\mathcal{D}$ and select a human-curated reference dataset $R_{\mathcal{D}}$ for each domain $\mathcal{D}$.

Furthermore, we need to define a set of characteristics that are indicative of data quality. We extend and adjust characteristics found in previous works on dataset generation to evaluate a given synthetic dataset $S_{\mathcal{D}}$ using five important characteristics: faithfulness, diversity, conformity, complexity, and downstream performance. Other than performance, of these metrics, faithfulness and conformity have been used in previous work directly, though not as the main focus of their evaluation (Ye et al., 2022a;b). Additionally, we modify the diversity metric used in these works to be suitable for our purposes and introduce complexity as a new characteristic to provide a full and comprehensive evaluation of a synthetic dataset. We now describe each of these characteristics in detail.

**Faithfulness** We start by considering faithfulness, i.e., how well a dataset fits the given domain $\mathcal{D}$. Faithfulness quantifies how much how much noise is introduced by the generation process that may impair model training. To measure faithfulness, we fine-tune a small classifier $M_{R_{\mathcal{D}}}$ on the reference dataset $R_{\mathcal{D}}$, and evaluate its performance in terms of accuracy on the respective synthetic dataset $S_{\mathcal{D}}$. We thus measure faithfulness as

$$\text{faithfulness}(S_{\mathcal{D}}) = \text{accuracy}(M_{R_{\mathcal{D}}}, S_{\mathcal{D}}).$$

We note that faithfulness does not only measure the correctness of labels associated with generated samples. It is also influenced by the quality of the generated samples themselves and whether they are representative of the reference dataset since non-representative samples are more likely to be misclassified by the classifier. Furthermore, while the classifier can provide an estimate of the faithfulness of the dataset, it is not a perfect measure and may be influenced by the quality of the classifier. However, since the classifier is the same for all synthetic datasets, we can still use it to compare the faithfulness of different datasets.

**Diversity** While LMs may generate faithful datasets, we need to ensure that the resulting samples are diverse rather than repetitive. To account for this, we also measure diversity, i.e., how distinct individual samples in the dataset are. Previous work on text dataset generation rely on Self-BLEU (Zhu et al., 2018) or Distinctness-$n$ (Li et al., 2016) to measure diversity. However, these metrics are not suitable for the purposes of evaluating the diversity of the text generated by LMs. Indeed, Self-BLEU and Distinctness-$n$ exhibit a logarithmic dependence on dataset size as demonstrated in App. A. Therefore, these metrics are not directly comparable across different datasets and cannot be used to evaluate the inherent diversity of an LMs within a given domain.

We therefore propose a normalized version of Distinctness-$n$ to correct for its size dependence. We do so by averaging Distinctness-$n$ over random subsets of the dataset of fixed size $k$. By keeping $k$ constant throughout all experiments, this metric is directly comparable across different datasets.

More concretely, given a dataset $S_{\mathcal{D}}$, let $L(S_{\mathcal{D}})$ be the multi-set obtained by lemmatizing all samples in $S_{\mathcal{D}}$ and collecting the obtained words. Let $C(X)$ denote the unique number of tokens among $X = x_1, ..., x_k$. We define the diversity as

$$\text{diversity}_k(S_{\mathcal{D}}) = \frac{1}{k} \mathbb{E}_X \left[ C(x_1, ..., x_k) \right]$$
$$\text{where } X \subseteq L(S_{\mathcal{D}}), |X| = k$$

**Conformity** While text generated with recent iterations of instruction-tuned models Chiang et al. (2023); OpenAI (2023); Geng et al. (2023) can be of high quality, diverse and faithful, a resulting dataset may still not fit the distribution of human-written text in a more casual setting due to the inability of these models to generate human-like text. Since common data domains contain a lot of internet-based dialect, overly high-quality responses may fall out of distribution. For example, a dataset for movie review analysis may contain both positive and negative reviews, but overall writing skills per author may vary. If a corresponding synthetic dataset only contains high-quality reviews, it may not be representative of the real distribution of reviews.

To capture this, we measure *conformity* to quantify the similarity between the distributions of a synthetic dataset and a real reference dataset. For this, we employ the MAUVE metric Pillutla et al. (2021), which indicates differences between two text distributions by calculating the Kullback-Leibler (KL) divergence between their smoothed representation in sentence embedding space.

$$\text{conformity}(S_{\mathcal{D}}) = \text{mauve}(R_{\mathcal{D}}, S_{\mathcal{D}})$$

**Complexity** Furthermore, it is possible that synthetic data looks natural and diverse, but the resulting samples are overly simplistic, e.g. when synthetic positive movie reviews only consist of reviews that are very good without any nuance in the samples. A classifier trained on an overly simplistic dataset has worse generalization error and therefore less utility. We therefore include the degree of data *complexity* as a core characteristic. To measure this, we train a small classifier $M_{train(S_{\mathcal{D}})}$ on a training split of the synthetic dataset $S_{\mathcal{D}}$ under consideration, and evaluate its accuracy on a held-out (also synthetic) validation split. Based on this, we define the complexity inversely proportional to the resulting validation accuracy, as follows:

$$\text{complexity}(S_{\mathcal{D}}) = 1 - \text{accuracy}(M_{train(S_{\mathcal{D}})}, val(S_{\mathcal{D}}))$$

Table 1: Overview of the models and training regimes considered in our comparative analysis.

| | Vanilla | | | | | Instruction-Tuned | | | | |
|---|---|---|---|---|---|---|---|---|---|---|
| | **350M** | **1.2B** | **6-7B** | **13B** | **175B** | **350M** | **1.2B** | **6-7B** | **13B** | **175B** |
| GPT-3 | ✓ | ✓ | ✓ | | ✓ | ✓ | ✓ | ✓ | | ✓ |
| GPT-3.5 | | | | | | | | | | ✓✓✓[†] |
| Falcon | | | ✓ | ✓ | | | | | | |
| Llama-1 | | | ✓ | ✓ | | | | | ✓[‡] | |
| Llama-2 | | | ✓ | ✓ | | | | ✓ | ✓ | |
| CodeLlama | | | ✓ | | | | | | | |

[†] INSTRUCTGPT-3.5-175B$_{\text{PPO}}$, INSTRUCTGPT-3.5-175B$_{\text{chat}}$ and INSTRUCTGPT-3.5-175B$_{\text{chat-instruct}}$
[‡] We use Vicuna-$7B, 13B$ as instruction-tuned Llama-1 models.

This metric allows us to measure the complexity of a dataset by measuring the generalization error on the same distribution. If this error is very low, the dataset is overly simplistic and the model can likely not generalize well to samples from the unseen reference dataset and has therefore low utility.

**Performance**   Overall, the four previous characteristics are summarized in the *performance* of a synthetic dataset. The generalization performance is measured by training a model on the synthetic dataset $S_{\mathcal{D}}$ and evaluating it on the reference dataset $R_{\mathcal{D}}$. We therefore report the accuracy of a model $M_{S_{\mathcal{D}}}$ trained on $S_{\mathcal{D}}$ and evaluated on the reference dataset $R_{\mathcal{D}}$ as the metric

$$\text{performance}(S_{\mathcal{D}}) = \text{accuracy}(M_{S_{\mathcal{D}}}, R_{\mathcal{D}}).$$

## 4   EVALUATION

To assess the generative abilities of different models, we apply our evaluation framework to a wide range of different models belonging to five different size classes ($350M - 175B$), four different families (*GPT, Falcon, Llama-1 and LLama-2*) and two different training regimes (vanilla, i.e., non-instruction-tuned, and instruction-tuned).

In this section, we first describe our experimental setup and then discuss our main results through two lenses: (1) We identify three inherent tradeoffs between our core characteristics, that we observe consistently across all models, and, (2) we compare the different models and training regimes in terms of their generative performance, as measured by our framework.

**Data Domains**   We choose common data domains that we can safely assume to be in-distribution for all examined models, and that we can find real-world reference datasets for. Concretely, we use AGNews (Zhang et al., 2015) to perform news genre classification for news headlines, SST-2 (Socher et al., 2013) for sentiment analysis of movie reviews, ELI5 for subreddit classification of online forum posts (Fan et al., 2019) and a subset of GoEmotions (Demszky et al., 2020) for general emotion classification. For more details on these domains and the reference datasets, we refer to App. B.

**Models and Prompting**   We provide an overview of all models (Almazrouei et al., 2023; Touvron et al., 2023a;b; Chiang et al., 2023; Brown et al., 2020) and training regimes (Ouyang et al., 2022) considered in our analysis in Table 1. For data generation, we mostly rely on simple zero-shot instructive prompts to generate data for a given domain, allowing us to directly access the raw distribution modeled by a given model. For the classifiers trained as part of our evaluation framework, we fine-tune pre-trained DistilBERT (Sanh et al., 2019) models. For further details we refer to App. C.

**Sampling and Aggregation**   For each domain and model, we generate synthetic datasets of 3000 samples each. We do so for up to 5 different sampling temperatures $T \in \{0.7, 1.0, 1.3, 1.6, 1.9\}$ for instruction-tuned models and $T \in \{0.7, 1.0, 1.3\}$ for vanilla models as samples quickly become degenerate. For most models, we also include a sample from the nucleus distribution with $p = 0.9$.

To account for the stochasticity of the sampling process, we generate 5 datasets per configuration and report the average results for our metrics across these 5 datasets. In App. F, we discuss the resulting standard deviations, which are small and do not impact the conclusions presented here.

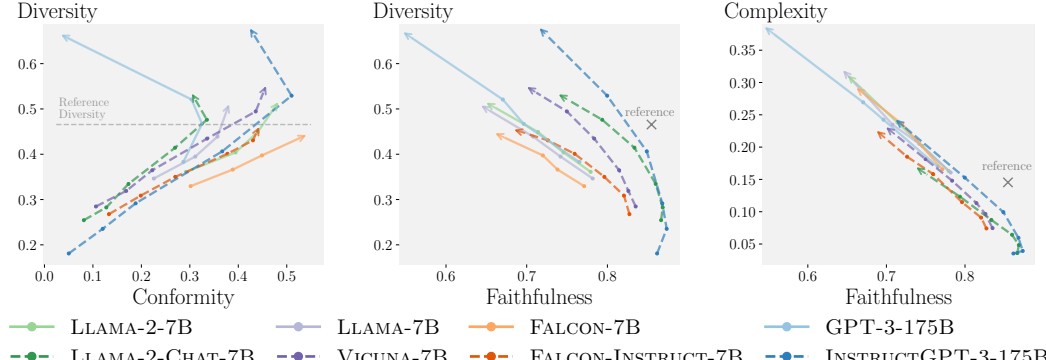

Figure 2: Tradeoffs between various metrics in the zero-shot setting. From left to right: Tradeoffs in diversity and conformity, diversity and faithfulness, and complexity and faithfulness. Arrows indicate the direction of higher sampling temperature for the same model.

In our main evaluation, we always report the average of our metrics across all considered data domains, and typically focus only on a subset of models, to facilitate readability. Still, our results hold for all considered models and domains, unless otherwise noted. We include full results in App. E.

## 4.1 MODEL-INHERENT CHARACTERISTICS

We first look at model-inherent characteristics by considering tradeoffs between our core characteristics, sampling from the same model with varying temperature. Naturally, higher sampling temperature can be expected to correlate with diversity, however, we further observe other meaningful tradeoffs. To illustrate, Fig. 2 shows the underlying relationships of our characteristics, plotting diversity, conformity, faithfulness and complexity, where directed arrows indicate the effect of increasing sampling temperature with a given model. We now discuss each of these plots in turn. In App. D, we show that the same tradeoffs hold when the dependent variable is model size.

**1. Conformity v. Diversity** We find that as conformity increases, diversity increases (Fig. 2, left). However, as soon as a threshold in reference diversity is reached, this trends reverses. We explain this by that fact that conformity actually measures closeness of the synthetic data distribution to the real reference distribution. At the same time, diversity can be seen as a measure for how wide this distribution is. When the width (or diversity) of the reference and synthetic dataset match, conformity will generally be higher. However, lower diversity means that the resulting dataset provides a very narrow view on the data domain, while higher diversity results in a dataset that represents concepts from outside the data domain as well. More technically, we observe a quadratic dependence of conformity and diversity, which is statistically significant (p-value $< 0.001$) and centered around a diversity of $0.5$, slightly higher than the average diversity of the reference dataset.

**2. Diversity v. Faithfulness** We observe an inverse linear relationship between diversity and faithfulness (Fig. 2, middle), as faithfulness decreases with increasing diversity. This is because higher diversity indicates samples from a wider distribution which generally also includes samples atypical for the domain and reference dataset, i.e. samples that are not faithful to the reference dataset. Here, the difference between instruction-tuned (dashed lines) and vanilla (full lines) models is especially notable. While vanilla models generate more diverse datasets for the same temperature, for a fixed level of faithfulness, instruction-tuned models generate more diverse datasets.

**3. Faithfulness v. Complexity** Finally, we observe a strong linear relationship between faithfulness and complexity (Fig. 2, right), with a Pearson correlation coefficient of $-0.93$. In an ideal scenario, a classification model trained on the generated dataset is equal to the model trained on the reference datasets. In such a case, faithfulness would equal $1 -$ complexity. However, as shown in the figure, we find that models appear shifted with respect to each other and do not follow the inverse relationship perfectly. This suggests the existence of a model-inherent faithfulness-complexity ratio, which in turn is an important indicator of dataset quality. In fact, further linear analysis reveals that the sum of the faithfulness and complexity metrics is an equally good predictor of dataset performance as the individual metrics, showing that the main dependence of performance is captured by their sum.

Table 2: Comparing LLAMA-based and LLAMA-2-based model for sampling temperature $T = 1$ in the zero-shot setting. Metrics for real data are measured with respect to a held-out validation set.

| Model Name | Complexity | Faithfulness | Diversity | Conformity | Performance |
|---|---|---|---|---|---|
| Real data | 0.145 | 0.855 | 0.466 | 0.963 | 0.855 |
| LLAMA-7B | 0.235 | 0.708 | 0.439 | 0.357 | 0.749 |
| LLAMA-2-7B | **0.238** | **0.714** | **0.449** | **0.440** | **0.754** |
| VICUNA-7B | **0.113** | 0.815 | **0.365** | **0.222** | 0.744 |
| LLAMA-2-CHAT-7B | 0.065 | **0.860** | 0.334 | 0.173 | **0.749** |

## 4.2 MODEL COMPARISON

Going beyond model-inherent characteristics, we now compare *across* different models and training regimes in terms of our evaluation metrics and the identified tradeoffs from the previous section. We first discuss the effect of instruction-tuning and then compare the different model families. Finally, we compare the different models in terms of their overall performance.

**Instruction-tuning**   Firstly, we observe a clear difference between instruction-tuned and vanilla models. While neither consistently balances all metrics and tradeoffs, we generally find that instruction-tuned models are substantially more faithful, but exhibit less diversity than vanilla models (see Fig. 2, middle). Only at the highest sampling temperatures ($T = 1.9$) instruction-tuned models achieve similar faithfulness levels as their vanilla counterparts at the lowest temperatures ($T = 0.7$).

As shown in Fig. 2 (left), for LLAMA-$\{1, 2\}$ and FALCON we can observe that instruction-tuned models generally exhibit less conformity, compared to their vanilla variants. We explain this with the high levels of curation with instruction-tuning datasets, whereas some of the reference datasets contain ungrammatical or otherwise malformed samples. Interestingly, for OpenAI instruction-tuned models like INSTRUCTGPT-3-175B we observe the opposite behavior, even when considering OpenAI models of the same size as the LLAMA and FALCON variants. We show a full comparison of all models in App. E.

Considering complexity, we find that both vanilla and instruction-tuned synthetic datasets tend to be more complex at higher sampling temperatures (see Fig. 2 right). However, instruction-tuned models behave significantly worse in the faithfulness-complexity-tradeoff, i.e. achieve lower faithfulness at similar complexity levels. Again, OpenAI models like INSTRUCTGPT-3-175B appear to defy this and can match the vanilla models in this regard.

For downstream performance, we generally observe lower scores for instruction-tuned LLAMA-$\{1, 2\}$ and FALCON models, compared to their vanilla counterparts. However, for some of OpenAI's models the instruction-tuning process appears to actually enhance generative abilities with respect to our metrics, and thus also downstream performance. Overall, our metrics characterize the generative abilities of instruction-tuned OpenAI models very differently from comparable LLAMA and FALCON variants. This leads us to believe that OpenAI's concrete and proprietary instruction-tuning process and dataset must be substantially different from the ones used for LLAMA and FALCON.

**Comparing Model Families**   In Table 2, we compare LLAMA-7B and LLAMA-2-7B. We find that conformity serves as the primary indicator for differentiating LLAMA vanilla models of different generations, with LLAMA-2-7B showing significantly improved results.

With respect to instruction-tuning, we analyze the difference between VICUNA-7B and LLAMA-2-CHAT-7B. We find that in pure generative abilities according to our framework, VICUNA-7B outperforms LLAMA-2-CHAT-7B across almost all metrics, with the exception of faithfulness. Notably, VICUNA-7B only fine-tunes the model on dataset of instruction prompts, whereas the fine-tuning process for LLAMA-2-CHAT-7B is more extensive Touvron et al. (2023b), using multiple phases of fine-tuning and RLHF, similar to OpenAI's instruction-tuning process Ouyang et al. (2022). This suggests that the more extensive fine-tuning process of LLAMA-2-CHAT-7B does not necessarily lead to better generative abilities, at least not according to our metrics or downstream performance. We show the results for other temperatures in App. E, but note that the conclusions and trends discussed here do not change.

Table 3: Comparing models for the best performing temperature in the zero-shot setting. T is the optimal sampling temperature.

| Model Name | T | Perf. |
|---|---|---|
| INSTRUCTGPT-3-175B | 1.9 | 0.767 |
| LLAMA-2-7B | 0.7 | 0.760 |
| LLAMA-13B | 0.7 | 0.758 |
| VICUNA-7B | 1.9 | 0.757 |
| LLAMA-2-CHAT-7B | 1.6 | 0.755 |
| ⋮ | ⋮ | ⋮ |
| INSTRUCTGPT-3.5-175B$_{chat}$ | 1.9 | 0.715 |
| GPT-3-350M | 1.0 | 0.707 |
| INSTRUCTGPT-3-350M | 1.3 | 0.707 |

Table 4: Comparing models for the best performing temperature in the few-shot setting. T is the optimal sampling temperature.

| Model Name | T | Perf. |
|---|---|---|
| LLAMA-2-CHAT-13B | 1.6 | 0.775 |
| INSTRUCTGPT-3-175B | 1.3 | 0.775 |
| VICUNA-13B | 1.6 | 0.768 |
| LLAMA-2-CHAT-7B | 1.6 | 0.764 |
| VICUNA-7B | 1.6 | 0.764 |
| ⋮ | ⋮ | ⋮ |
| INSTRUCTGPT-3.5-175B$_{chat-instruct}$ | 1.3 | 0.744 |
| INSTRUCTGPT-3-350M | 0.7 | 0.723 |
| INSTRUCTGPT-3.5-175B$_{chat}$ | 1.3 | 0.711 |

**Model Comparison** To compare downstream performance independent from sampling configuration, we consider maximum downstream performance per model, choosing the best sampling temperature individually. We report the summary of the resulting ranking in Table 3 with full results in App. E. Interestingly, among the top positions we see both vanilla (LLAMA-2-7B, LLAMA-13B) and instruction-tuned models, suggesting that instruction-tuning does not necessarily enhance inherent generative capabilities. Further, we find that specifically INSTRUCTGPT-3.5-175B$_{chat}$ model scores very poorly on all metrics, including downstream performance. On closer look we find that its training regime appears to primarily optimize faithfulness at the expense of other distributional characteristics. This results in very poor downstream performance, only slightly better than the worst performing models GPT-3-350M and INSTRUCTGPT-3-350M, which are also much older than INSTRUCTGPT-3.5-175B$_{chat}$. Surprisingly, we find that the best model INSTRUCTGPT-3-175B is closely followed by much smaller LLAMA-based models, suggesting that a large model size is not necessarily a requirement for good generative abilities in a distributional sense.

**Few-Shot Performance** While in most of our experiments we rely on simple instructive prompts, we also consider few-shot prompting. Specifically, we select 10 samples from each data domain and use three random samples from those 10 samples for each sample query. With this, we can significantly boost the performance of instruction-tuned models specifically, as the additional variation and specificity during prompting helps address their low diversity and conformity scores. We report the updated ranking of downstream performance in Table 4. With this, instruction-tuned models dominate the top positions, with INSTRUCTGPT-3-175B and LLAMA-2-CHAT-13B sharing the first place. Interestingly, we find that vanilla models are now completely absent from the top five, indicating that the few-shot procedure is effective at mitigating the issue of instruction-tuned models regarding diversity and conformity. INSTRUCTGPT-3.5-175B$_{chat}$ on the other hand, moves to the last position, suggesting that even few-shot prompting cannot address the short-coming of `chat`-training for synthetic dataset generation.

## 5 RELATED WORK

Synthetic data generation using LLMs has been explored for various applications and use-cases. We briefly discuss each of these research areas.

**Dataset generation for zero-shot learning** Recent works (Ye et al., 2022a;b; Gao et al., 2023; Meng et al., 2022) have proposed alternative strategies for zero-shot learning due to the increasing size of foundation models. These works adopt a different paradigm that leverages large language models (LLMs) to generate synthetic datasets and train smaller, task-specific models on these datasets for downstream tasks. Ye et al. (2022a) pioneers this approach by introducing ZEROGEN, a framework that generates synthetic data using LLMs for various downstream tasks. Following this initial work, several studies have focused on improving the performance of models trained on synthetic data by addressing potential issues such as fitting to noisy samples and enhancing generalization to real-world applications (Gao et al., 2023; Meng et al., 2022). Additionally, Ye et al. (2022b) proposes an iterative

few-shot prompting method that incorporates influential samples from the synthetic data to increase dataset size, further refining the data generation process. Despite these advances, current studies do not consider larger foundation models and neglect to analyze the trade-offs and relationships between the different characteristics of a dataset.

Several works generate datasets for specific tasks. Schick and Schütze (2021) uses an instruction combined with an input sentence and uses a LM to generate a new sample, where the label relationship between the original and generated sentence can be derived from the instruction. Josifoski et al. (2023) generates data for complex tasks, such as structured data, by leveraging asymmetry in tasks.

**Self-improvement**   Using synthetic data to self-improve the foundation model has achieved a lot of attention recently. In particular, Bai et al. (2022b) introduced a novel method for training models to be more helpful using self-improvement by allowing the trained model to both generate and evaluate its own outputs and therefore iteratively improving its own performance. Similarly, Wang et al. (2022b) developed an approach wherein the model generates its own instruction dataset, which is then used for fine-tuning itself. Huang et al. (2023) focused on enhancing the model's capability in reasoning tasks by training it on its own high-confidence outputs. Taking a different approach, Haluptzok et al. (2023) aimed to enhance the code generation capabilities of the model. By employing the model's own output, they generated and selected code snippets to use training samples, ultimately improving the model performance in code-related tasks. Finally, to reduce the toxicity of AI-generated text, Wang et al. (2022a) proposed a method to fine-tune language models on non-toxic data.

**Dataset Augmentation**   Dataset augmentation has a rich history, with various strategies employed to improve the performance of models. Techniques such as back-translation Sennrich et al. (2016), c-BERT word replacement Wu et al. (2019), or a combination of different methods Qu et al. (2021) have been explored. Recently, LMs have also been used for data augmentation. For instance, Yang et al. (2020) generates samples using foundation models for commonsense reasoning tasks and incorporates the most diverse and informative samples into their dataset. Moreover, Dorner et al. (2023) utilizes foundation models for unsupervised style transfer.

Chia et al. (2022) generates synthetic data for relation triplet extraction, where the goal is to extract two parts of the prompt as well as their relation label. Bonifacio et al. (2022) generates data for an information retrieval task, but do require a few examples for each class.

**Generation for specific purpose**   Several works have focused on generating datasets for specific applications using foundation models. For instance, Chen et al. (2023) developed a dataset for social conversations, while Hartvigsen et al. (2022) introduced a new large-scale dataset for toxicity analysis. Additionally, Yuan et al. (2022) presented a human-in-the-loop dataset generation technique and employed it to create a dataset on biographies.

**LLM Evaluation**   A wide range of holistic multi-task multi-metric frameworks (Liang et al., 2022; Gao et al., 2021a; Hendrycks et al., 2021) as well as domain-specific evaluation suites (Guha et al., 2023) for the evaluation of LLMs have been proposed.These frameworks often build on existing tasks, such as question answering (Clark et al., 2018; Bhakthavatsalam et al., 2021; Lin et al., 2022), language understanding (Wang et al., 2019) or sentence completion (Zellers et al., 2019).While assessing models on a board set of downstream tasks, to the best of our knowledge, non of these works measure the models capacity for dataset generation.

## 6   CONCLUSION

Through a comprehensive evaluation of synthetic datasets generated by LLMs, our study revealed inherent tradeoffs between dataset diversity, complexity, conformity, faithfulness and performance. We show that these trade-offs generalize across data domains and models, allowing us to study differences between instruction-tuned and vanilla models. These results highlight differences in model characteristics, e.g., how different models in the LLAMA family differ. We further find that ChatGPT (INSTRUCTGPT-3.5-175B$_{chat}$) generates very faithful datasets, but lacks in all other models in terms of complexity, diversity and conformity resulting in a worse downstream performance compared to other models. Our study marks a crucial step towards a more nuanced understanding of dataset generation by LLMs, shedding light on the behavior of various models.

## 7 REPRODUCIBILITY

We include our code, prompts, and detailed instructions on how to reproduce our results as part of the supplementary material of this paper.

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

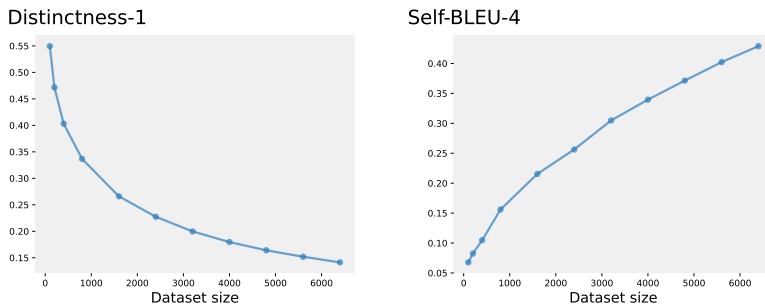

Figure 3: The dependence of Distinctness-$n$ and Self-BLEU on dataset size.

## A  SIZE-DEPENDENCE DIVERSITY

We show that Self-BLEU (Zhu et al., 2018) and Distinctness-$n$ (Li et al., 2016) metrics are dependent on the dataset size, making them unsuitable for our analysis. To illustrate this, we take our SST-2 dataset and truncate it at different sizes. A desirable property of any diversity metric is its consistency across these different sizes, as it enables us to assess the inherent diversity of the dataset generation process. However, Fig. 3 reveals that both Self-BLEU and Distinctness-$n$ do not exhibit this property. In particular, Self-BLEU increases as the dataset size grows, whereas Distinctness-$n$ decreases.

We note that both results are expected. Indeed, Distinctness-$n$, defined as the ratio of unique tokens to the total tokens, remains constant only if this proportion is maintained throughout the generation process. However, this constancy is improbable due to increased word repetitions with a larger number of previous words. Moreover, since Distinctness-$n$ relies on token count, it may differ even for two datasets of identical size but with varying generated sample lengths, deeming it an unsuitable metric for our study.

Self-BLEU's dependence on dataset size on the other hand stems from the fact that it is calculated as the mean over the BLEU scores of each sample with respect to all other samples. As the dataset size grows, the number of samples to compare to grows as well, and thus the mean BLEU score increases. Therefore, we conclude that Self-BLEU is also not a suitable metric for our analysis.

## B  DATASETS AND PROMPTS

In this section, we provide an overview of each data domain and reference dataset used in our study, along with the prompts designed for each domain.

**Movie reviews**  To investigate sentiment classification in the prevalent domain of movie reviews, we use the Stanford Sentiment Treebank (SST-2) dataset Socher et al. (2013). This dataset comprises movie reviews accompanied by binary sentiment labels (positive/negative). We create datasets specifically for movie sentiment analysis for this data domain.

**News Headlines**  To incorporate a domain demanding more world knowledge and generally characterized by more formal language, we generate datasets for news headline classification using the AGNews dataset Zhang et al. (2015). This dataset contains news headlines accompanied by brief descriptions, organized into four categories (Business, Sci/Tech, World, Sports). We use the news headlines from this dataset in our study and generate new ones with our procedure.

**Subreddits**  With the intention of including an informal task common in internet data analysis, we generate datasets for subreddit classification using the eli5 dataset Fan et al. (2019). This dataset consists of questions and answers posted on three distinct subreddits (AskScience, AskHistorians, and ExplainLikeImFive), where we use the questions of the dataset in this paper. We formulate questions that could potentially be posed in each of these subreddits.

Table 5: Prompts used for each task. Variables indicated by {class} are replaced by the respective class label. Variables indicated by {examples} are 3 example samples from the dataset. These samples are always selected from 10 random and fixed samples from the dataset.

| Task | Setting | Prompt |
|---|---|---|
| Movie Reviews | Standard | The movie review in positive {class} is: " |
| | Chat | Generate a very short {class} movie review. |
| | Few-Shot | The movie reviews in {class} sentiment are: {examples} \n 4. |
| | Chat Few-Shot | Generate one very short {class} movie review similar in style to the following ones: {examples} |
| News Headlines | Standard | The following news article title is in the category of '{class}': " |
| | Chat | Generate a news article title is in the category of '{class}'. |
| | Few-Shot | The following news article titles are in the category of '{class}': {examples} \n 4. |
| | Chat Few-Shot | Generate a news article title in the category of '{class}' similar in style to the following ones: {examples} |
| Subreddits | Standard | A question that appeared on the subreddit '{class}': " |
| | Chat | Generate a question that could appear on the subreddit '{class}'. |
| | Few-Shot | Questions that could appear on the subreddit '{class}': {examples} \n 4. |
| | Chat Few-Shot | Generate a question that could appear in the subreddit '{class}' in a similar in style to the following ones: {examples} |
| Emotions | Standard | The following reddit comment displays the emotion '{class}': " |
| | Chat | Generate a reddit comment that displays the emotion '{class}'. |
| | Few-Shot | The following reddit comments display the emotion '{class}': {examples} \n 4. |
| | Chat Few-Shot | Generate a reddit comment that displays the emotion '{class}' similar in style to the following ones: {examples} |

**Emotions**   Recognizing the significance of emotional language in various tasks like customer support, we create datasets for emotion classification using the GoEmotions dataset Demszky et al. (2020). This dataset features Reddit comments labeled with 27 distinct emotions. We limit our scope to 5 specific emotions (surprise, grief, nervousness, desire, gratitude) and generate datasets for emotion classification.

**Prompts**   We outline the prompts designed for all tasks in Table 5. While we use the same prompt for each model, we make an exception for INSTRUCTGPT-3.5-175B$_{\text{Chat}}$, which requires slightly different prompts to maintain comparability. For each dataset, we chose separate prompts for the normal and few-shot setups.

## C   EVALUATION DETAILS

**Training**   We fine-tune a DistilBERT Sanh et al. (2019) model for each experiment using a batch size of 8 and at most 5000 training steps or 5 epochs (whichever comes first). We use the AdamW optimizer Loshchilov and Hutter (2019) with a learning rate of $1e-5$. We use a learning rate scheduler with 600 warmup steps that decays the learning rate linearly to 0 after the warmup steps. Additionally, we also use temporal ensembling and label smoothing as regularization techniques as in Meng et al. (2022).

**Evaluation**   We fine-tune the model as described above on five datasets consisting of 3000 samples for each generated datapoints to measure complexity and performance. For faithfulness, we fine-tune the model on the real dataset and evaluate the accuracy of the model on each of the five generated datasets. Finally, to calculate diversity we set the hyperparameter in our definition to $k = 5000$ for our experiments.

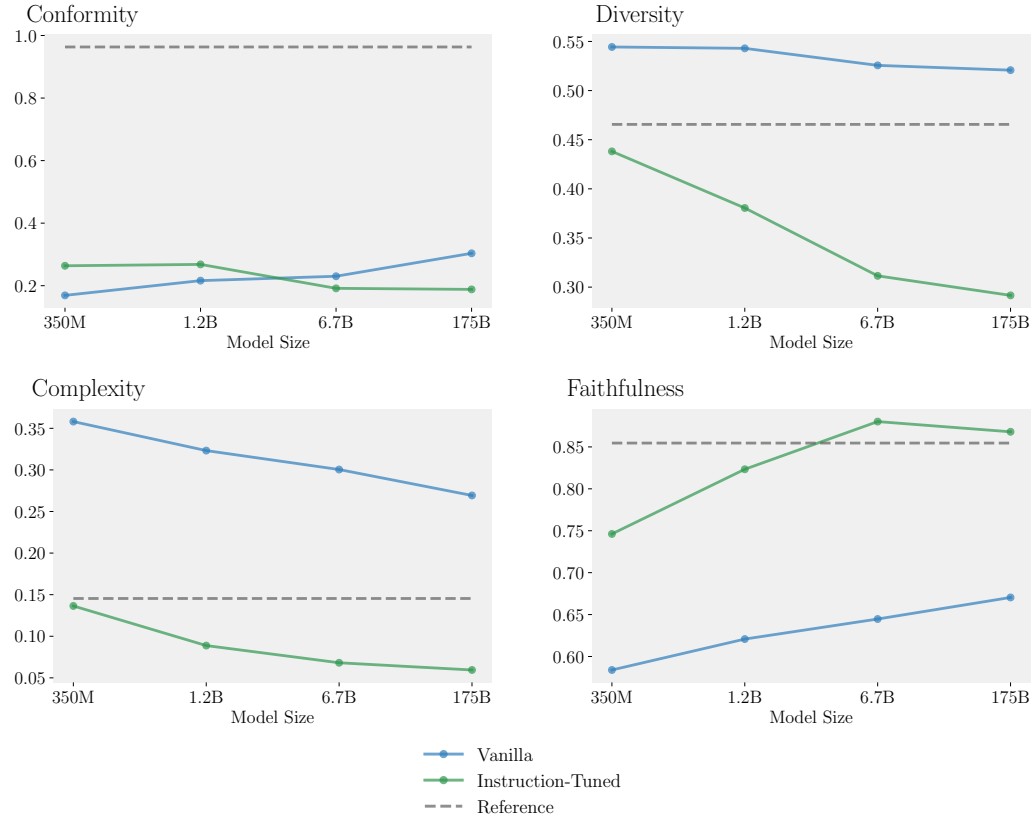

Figure 4: Metric values for all metrics in the zero-shot setting for the GPT-3 model family for sampling temperature $T = 1$. Results ordered by model size.

# D TRADEOFFS FOR MODEL SIZE

In §4.1, we explore tradeoffs in relation to model size within the GPT-3 model family, for both vanilla and instruction-tuned variants. The impact of these tradeoffs on key characteristics is illustrated in Fig. 4. Our findings show that the tradeoffs identified in §4.1 are also true when the dependent variable is model size. Specifically, faithfulness improves with size in both model types, probably because of their stronger capabilities. Conversely, both diversity and complexity drop, aligning with the tradeoffs outlined in §4.1.

When examining conformity in relation to model diversity, vanilla models, show an inverse relationship: as diversity decreases, conformity increases. However, in instruction-tuned models, a decrease in diversity leads to reduced conformity. This difference can be explained by the quadratic relationship between diversity and conformity. Specifically, as the diversity of vanilla models decreases, it converges towards the diversity seen in the reference data, thus increasing conformity. In contrast, for instruction-tuned models, a decrease in diversity results in a divergence from the reference data's diversity, thereby lowering conformity.

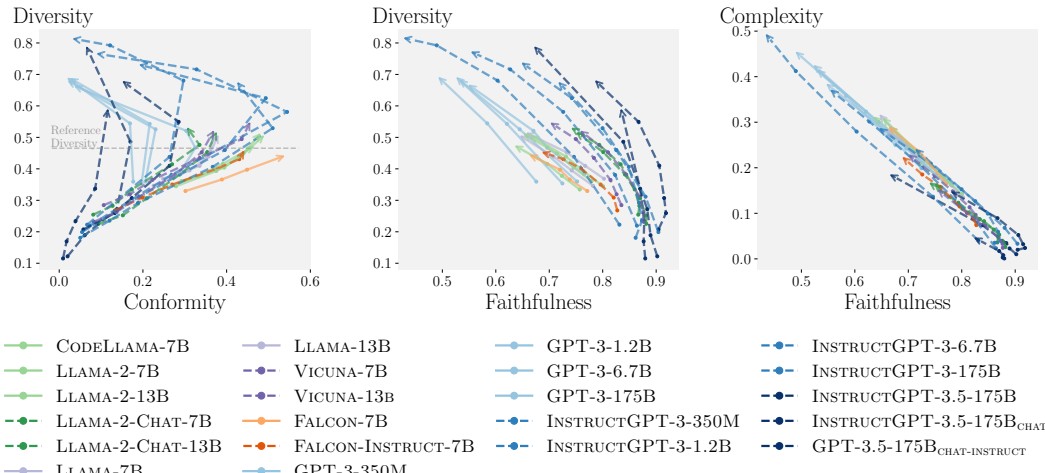

Figure 5: Tradeoffs between various metrics in the zero-shot setting. From left to right: Tradeoffs in diversity and conformity, diversity and faithfulness, and complexity and faithfulness. Arrows indicate the direction of higher sampling temperature for the same model. For vanilla models, temperatures range $0.7 - 1.3$ in steps of $0.3$ and for instruction-tuned models temperatures range between $0.7 - 1.9$. Additionally, for each open-source model and GPT-3-175B, INSTRUCTGPT-3-175B and INSTRUCTGPT-3.5-175B$_{\text{PPO}}$ one point is added using nucleus sampling with $p = 0.9$.

Table 6: Comparing LLAMA-based and LLAMA-2-based model for sampling temperature $T = 1$ in the zero-shot setting. Metrics for real data are measured with respect to a held-out validation set.

| Model Name | Temperature | Complexity | Faithfulness | Diversity | Conformity | Performance |
|---|---|---|---|---|---|---|
| Real data | - | 0.145 | 0.855 | 0.466 | 0.963 | 0.855 |
| LLAMA-7B | 0.7 | 0.161 | 0.782 | 0.346 | 0.226 | 0.745 |
| LLAMA-2-7B | 0.7 | 0.163 | 0.779 | 0.361 | 0.292 | 0.760 |
| LLAMA-7B | 1.0 | 0.235 | 0.708 | 0.439 | 0.357 | 0.749 |
| LLAMA-2-7B | 1.0 | 0.238 | 0.714 | 0.449 | 0.440 | 0.755 |
| LLAMA-7B | 1.3 | 0.305 | 0.655 | 0.495 | 0.376 | 0.745 |
| LLAMA-2-7B | 1.3 | 0.298 | 0.662 | 0.502 | 0.474 | 0.750 |
| LLAMA-2-CHAT-7B | 0.7 | 0.036 | 0.867 | 0.254 | 0.081 | 0.727 |
| VICUNA-7B | 0.7 | 0.075 | 0.835 | 0.285 | 0.106 | 0.735 |
| LLAMA-2-CHAT-7B | 1.0 | 0.064 | 0.860 | 0.334 | 0.173 | 0.749 |
| VICUNA-7B | 1.0 | 0.113 | 0.815 | 0.365 | 0.222 | 0.746 |
| LLAMA-2-CHAT-7B | 1.3 | 0.087 | 0.834 | 0.414 | 0.270 | 0.754 |
| VICUNA-7B | 1.3 | 0.148 | 0.784 | 0.435 | 0.336 | 0.755 |
| LLAMA-2-CHAT-7B | 1.6 | 0.123 | 0.794 | 0.476 | 0.335 | 0.755 |
| VICUNA-7B | 1.6 | 0.181 | 0.750 | 0.494 | 0.436 | 0.753 |
| LLAMA-2-CHAT-7B | 1.9 | 0.158 | 0.751 | 0.520 | 0.312 | 0.752 |
| VICUNA-7B | 1.9 | 0.219 | 0.712 | 0.536 | 0.452 | 0.756 |

## E  FULL RESULTS

We provide extra plots for the various tradeoffs between characteristics for all models averaged in Fig. 5 and for each dataset separately in Fig. 6.

Full tables for Table 2, Table 3 and Table 4 are provided in resp. Table 6, Table 7 and Table 8.

We also provide the metrics with respect to the reference datasets for all datasets separately in Table 9.

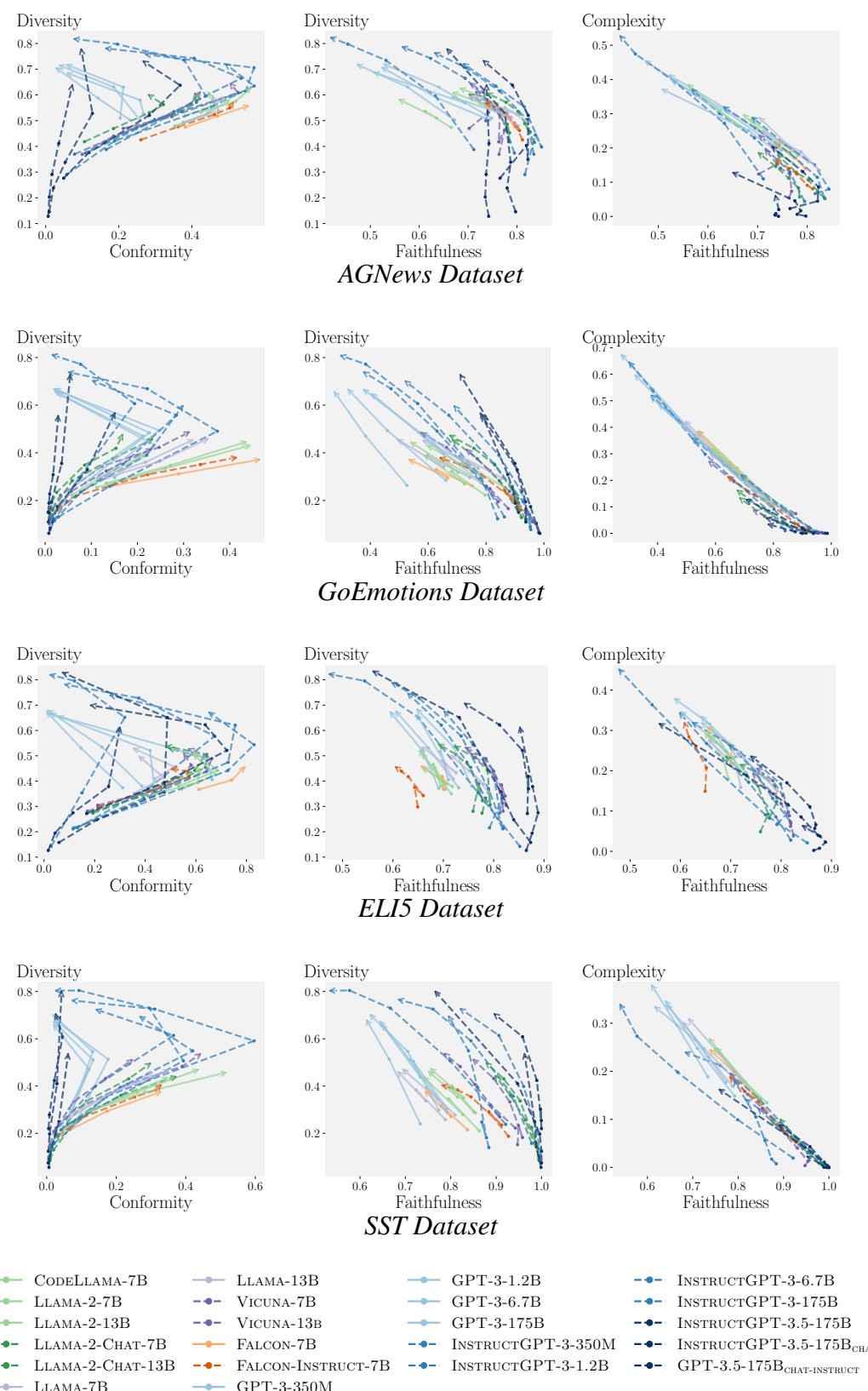

Figure 6: Tradeoffs between various metrics in the zero-shot setting. Arrows indicate the direction of higher sampling temperature for the same model. Sampling temperatures start at $0.7$ in spaces of $0.3$. Additionally, for each open-source model and GPT-3-175B, INSTRUCTGPT-3-175B and INSTRUCTGPT-3.5-175B$_{PPO}$ one point is added using nucleus sampling with $p = 0.9$.

Table 7: Comparing models for the best performing temperature in the zero-shot setting. T is the optimal sampling temperature.

| Model Name | Temperature | Performance |
|---|---|---|
| INSTRUCTGPT-3-175B | 1.9 | 0.767 |
| LLAMA-2-7B | 0.7 | 0.760 |
| LLAMA-13B | 0.7 | 0.758 |
| VICUNA-7B | 1.9 | 0.756 |
| LLAMA-2-CHAT-7B | 1.6 | 0.755 |
| FALCON7B | 1.0 | 0.754 |
| INSTRUCTGPT-3.5-175B$_{PPO}$ | 1.6 | 0.750 |
| LLAMA-7B | 1.0 | 0.749 |
| GPT-3-6.7B | 1.0 | 0.749 |
| LLAMA-2-13B | 1.3 | 0.749 |
| LLAMA-2-CHAT-13B | 1.9 | 0.748 |
| VICUNA-13B | 1.6 | 0.748 |
| INSTRUCTGPT-3-6.7B | 1.3 | 0.743 |
| GPT-3-175B | 0.7 | 0.737 |
| FALCON-INSTRUCT-7B | 1.0 | 0.733 |
| INSTRUCTGPT-3.5-175B$_{chat-instruct}$ | 1.9 | 0.732 |
| INSTRUCTGPT-3-1.2B | 1.9 | 0.728 |
| GPT-3-1.2B | 1.0 | 0.722 |
| CODELLAMA-7B | 0.7 | 0.722 |
| INSTRUCTGPT-3.5-175B$_{chat}$ | 1.9 | 0.715 |
| GPT-3-350M | 1.0 | 0.707 |
| INSTRUCTGPT-3-350M | 1.3 | 0.707 |

Table 8: Comparing models for the best performing temperature in the few-shot setting. T is the optimal sampling temperature.

| Model Name | Temperature | Performance |
|---|---|---|
| LLAMA-2-CHAT-13B | 1.6 | 0.775 |
| INSTRUCTGPT-3-175B | 1.3 | 0.775 |
| VICUNA-13B | 1.6 | 0.768 |
| LLAMA-2-CHAT-7B | 1.6 | 0.764 |
| VICUNA-7B | 1.6 | 0.764 |
| LLAMA-13B | 1.0 | 0.763 |
| INSTRUCTGPT-3.5-175B$_{PPO}$ | 1.0 | 0.762 |
| GPT-3-175B | 1.0 | 0.760 |
| LLAMA-2-13B | 1.0 | 0.759 |
| INSTRUCTGPT-3-6.7B | 1.0 | 0.758 |
| CODELLAMA-7B | 1.6 | 0.756 |
| LLAMA-2-7B | 1.0 | 0.751 |
| INSTRUCTGPT-3-1.2B | 1.3 | 0.746 |
| INSTRUCTGPT-3.5-175B$_{chat-instruct}$ | 1.3 | 0.744 |
| INSTRUCTGPT-3-350M | 0.7 | 0.723 |
| INSTRUCTGPT-3.5-175B$_{chat}$ | 1.3 | 0.711 |

Table 9: Metrics for reference datasets on all datasets separately.

| Dataset | Complexity | Faithfulness | Diversity | Conformity | Performance |
|---|---|---|---|---|---|
| AGNews | 0.142 | 0.858 | 0.560 | 0.959 | 0.858 |
| SST-2 | 0.093 | 0.907 | 0.382 | 0.959 | 0.904 |
| ELI5 | 0.163 | 0.838 | 0.522 | 0.960 | 0.838 |
| GoEmotions | 0.184 | 0.816 | 0.378 | 0.976 | 0.819 |

Table 10: Standard deviations of all metrics across all models.

| Characteristic | Maximum std | Mean std |
|---|---|---|
| Complexity | 0.0030 | 0.0008 |
| Conformity | 0.0077 | 0.0018 |
| Diversity | 0.0022 | 0.0006 |
| Faithfulness | 0.0008 | 0.0006 |
| Performance | 0.0193 | 0.0006 |

## F    STANDARD DEVIATION METRICS

We briefly analyze the standard deviation of the reported numbers in §4. The maximum and mean standard deviations for all characteristics across the generated datasets are displayed in Table 10. It is evident that the standard deviation is relatively low, especially when compared to the reported differences between models. The performance characteristic exhibits the highest standard deviation, but even in this case, the maximum value is above $0.01$, which is very low when compared to typical values ranging from $0.7$ to $0.8$.

