# OpenReview forum: "Understanding Large Language Models Through the Lens of Dataset Generation"
_ICLR.cc/2024/Conference — Submitted to ICLR 2024_

### Official Review · Reviewer_Afrd · 2023-10-12

**Soundness:** 2 fair
**Presentation:** 3 good
**Contribution:** 2 fair
**Rating:** 3
**Confidence:** 5

**Summary:**

This paper examines the generation of text datasets using Large Language Models (LLMs) with a focus on distributional metrics like data diversity and faithfulness. It reveals trade-offs between these metrics across different LLMs and training methods, highlighting the impact of popular instruction-tuning techniques on LLM text generation abilities.

**Strengths:**

1. The studied task on using LLMs for data generation is interesting and can be useful for the research community.

2. The authors conduct experiments on various datasets and LLMs (including both open-sourced and close-sourced models).

3. The paper is overall easy to read.

**Weaknesses:**

1. The authors only consider the most simple prompts for the target tasks. However, there are several works that aim to improve the quality of prompts to yield higher-quality datasets, some examples include:

- Chung et al. "Increasing Diversity While Maintaining Accuracy: Text Data Generation with Large Language Models and Human Interventions." ACL 2023.

- Yu et al. "Large language model as attributed training data generator: A tale of diversity and bias." NeurIPS D&B Track, 2023.

It is also important to note that some dimensions (e.g. diversity) have already been studied in this work. As a result, some of the conclusions in this paper are already known and there are not many new insights about using LLMs for data generation.

2. Unsupported Claims. The paper raises a claim that "reinforcement learning with human feedback (RLHF) in ChatGPT leads to a significant degradation in synthetic dataset generation capabilities." However, the paper lacks a clear explanation of how the authors attribute this performance drop specifically to RLHF. A more detailed description of the experimental setup and results related to this assertion would enhance the paper's clarity and credibility.

3. In the main paper, the author only considers the average performance over different patterns, which can be less informative as different datasets show diverse patterns (according to Figure 5).

4. For the metrics, it is somehow not clear why using `unique number of tokens` as the metrics of Diversity.

**Questions:**

1. Could you elaborate on why this paper primarily relies on simple prompts for target tasks, especially when recent research has emphasized advanced prompt engineering techniques for improving dataset quality? How might incorporating more sophisticated prompts affect the study's outcomes?

2. Given that some dimensions, like diversity, have already been studied in this work, what new insights or contributions does this paper bring to the field of using LLMs for data generation?

3. In the paper, you assert that "reinforcement learning with human feedback (RLHF) in ChatGPT leads to a significant degradation in synthetic dataset generation capabilities." Could you provide a more detailed explanation of the experimental design and results that support this claim?

4. What conclusions can be made after your study? What are the recommendations for practitioners to use LLMs for training data generation? Currently, it is not very clear after reading this paper, so I feel readers will not benefit much from this paper.

---

> ### Author Response · Authors · 2023-11-21
>
> We thank the reviewer for their insightful questions, which we address below. We hope that our reply will be able to address their reservations and lead to a more positive view on our work.
>
> **Q: Could you elaborate on why this paper primarily relies on simple prompts over sophisticated prompt engineering?**
>
> The goal of our paper is to analyze the behavior and innate text generation capabilities of language models. We believe dataset generation provides an alternative to the more standard perplexity and loss measures for this capability, as it enables us to investigate how well the language model can model common distributions present in the training data rather than focusing on individual samples. The purpose of the prompt is therefore only to guide the language model towards the distribution of the data domain under investigation. While more advanced prompt engineering would create the possibility of generating datasets with a higher performance, it would also make it impossible to compare vanilla models with their instruction-tuned variant since prompt engineering techniques work differently with instruction-tuned models and it would therefore not allow us to measure the distributional shift between these models. To investigate more foundational characteristics, we thus focus on the relatively unprompted, direct model distribution, not a highly prompt-conditioned, specific model variant.
>
> **Q: Why are you using a unique number of tokens as the metric diversity?**
>
>
> We experimented with several metrics to see what the best metric for measuring diversity was. Two of those metrics were Self-BLEU (Zhu et al., 2018) and average pairwise sentence similarity (e.g., used in Yu et al., 2023). We found that Self-BLEU was highly correlated with the unnormalized version of our diversity metric, but took a lot longer to compute. Pairwise sentence similarity did not seem to capture diversity of the entire dataset as it has several modes of collapse based on the fact that it only looks at a dataset by comparing two sentences at a time. One simple example is a dataset that contains only two unique sentences which are repeated. In this case, Self-BLEU and our used metric correctly indicate a diversity close to 0, while average sentence similarity will measure a very high diversity if the two sentences are dissimilar.
>
> **Q:  Given that some dimensions, like diversity, have already been studied in previous work, what are your novel insights or contributions?**
>
> In contrast to prior work, which uses dataset generation to get optimal performance for a downstream task, we use it as a tool to evaluate language models and compare the differences between them. Our framework allows us to evaluate the models along five dimensions and compare a wide range of models in terms of innate dataset generation capabilities.
>
> We find and discuss several tradeoffs between the evaluated dimensions and analyze the differences between model families and training paradigm. For example, the fact that Llama-2 only increases performance with respect to Llama on the conformity metric, indicates that while Llama-2 has a better understanding of generated text and can generate samples more similar to our reference datasets, it does not improve its understanding of the actual outputs, since its faithfulness remains the same compared to Llama.
>
> Further, we find that there are significant differences between vanilla models and instruction-tuned variants. We show that while instruction-tuned models are more faithful, a trait they are specifically trained for, they exhibit less conformity and diversity, indicating that this training paradigm might not be optimal for dataset generation. We can explain this by noting that fine-tuning happens on specific tasks and instructions which are not indicative of data usually found on the internet. Therefore, the fine-tuning stage biases the model to output data that is more formal and grammatically correct than what appears in human-generated datasets.

---

> > ### Author Response · Authors · 2023-11-21
> >
> > **Q: Can you provide a more detailed explanation of the experimental design and results that support RLHF negatively impacting ChatGPTs text generation capabilities?**
> >
> > When comparing ChatGPT (InstructGPT-3.5-175B$_\text{chat}$) with all other models evaluated in the study, we find that the downstream performance of datasets generated by ChatGPT is among the lowest, only the smallest models exhibit a worse performance. This fact shows that the innate dataset generation capabilities for ChatGPT are worse than for other models. When comparing the various metrics, we find that ChatGPT generates incredibly faithful datasets, but these datasets are also very simple, not diverse and non-conform. Since the RLHF phase optimizes the model to answer questions politely, correctly and without grammatical mistakes, it only generates data that abides by these styles. While one could construct more complicated prompts to adjust for this, it would require the user to convey the distribution of the data domains in specific instructions which is error prone and difficult.
> >
> > In order to clarify this point, we changed our statement to “We further find that ChatGPT (InstructGPT-3.5-175B$_\text{chat}$) generates very faithful datasets, but lacks in all other models in terms of complexity, diversity and conformity resulting in a worse downstream performance compared to other models.”
> >
> >
> > **Q: What are the recommendations for practitioners to use LLMs for training data generation?**
> >
> > Please see Q1 of our main reply.
> >
> > We hope to have been able to address all the reviewers’ concerns, are happy to answer any follow-up questions they might have, and are looking forward to their reply.

---

> ### Comment · Reviewer_Afrd · 2023-11-21
> **Thank you for the detailed response**
>
> The reviewer appreciates the author's response. While I appreciate the explanation provided, I find it somewhat unconvincing for several reasons:
> - I have taken a look at Table 5, and found the prompts used for different models are not the same. This may introduce additional noise in evaluation. Besides, the authors only conducted experiments with a single set of prompt templates/verbalizer, which made the evaluation less convincing, as these templates can have a huge impact on the quality of the generated dataset.
> - For the prompt engineering techniques, the response underestimates the potential of sophisticated prompt engineering. Advanced prompts can be designed to guide models towards specific data distributions. The argument that advanced prompts would make comparisons impossible seems overly simplistic.
> - About the diversity metric, maybe my understanding is wrong, but this metric seems to be biased toward longer sequences. Besides, the faithfulness, conformity, and complexity all rely on additional language models for calculating the results. Not sure if the result is robust to selection of the model.
> - For the discussion of the insights, I feel the claim “We show that while instruction-tuned models are more faithful, a trait they are specifically trained for, they exhibit less conformity and diversity, indicating that this training paradigm might not be optimal for dataset generation.” Is overclaimed and may not be true in practice. From the experiment you run, the only conclusion you can draw is “the instruction-tuned model does not work well with the specific prompt used in the experiments”.  For instance, instruction-tuned models may excel with more complex prompts and benchmarks like GLUE/SuperGLUE, which should be considered in a broader assessment. It is not appropriate to dismiss other models without providing comprehensive results.
> - The recommendation for practitioners seems to be very hand-waving and not “practical” at all. First, the statement, "While ChatGPT is a popular choice, it may not always be the most suitable," lacks specificity and could apply to any model (e.g., LLama2, Vicuna). A more informative statement might be, "While ChatGPT is a popular choice, it may not always be the most suitable when considering the simple prompts used in our paper." Second, the claim that “a combination of different LLMs could increase performance and mitigate problems or biases related to a specific model” is also not informative. Is there any result in the paper about the performance of mixing these different generated data together? Third, the recommendation to "test different sampling temperatures for optimal performance" has been previously mentioned in [Chung et al., 2023], and it may not be practical to heavily tune this parameter within the strict few-shot/zero-shot setting [Bragg et al., 2021]. Therefore, it may be worth reconsidering the practicality and impact of this recommendation in real-world scenarios.
>
>
> Reference:
>
> John Chung, Ece Kamar, and Saleema Amershi. 2023. Increasing Diversity While Maintaining Accuracy: Text Data Generation with Large Language Models and Human Interventions. In Proceedings of the 61st Annual Meeting of the Association for Computational Linguistics (Volume 1: Long Papers), pages 575–593, Toronto, Canada. Association for Computational Linguistics.
>
> Bragg, Jonathan, et al. "Flex: Unifying evaluation for few-shot nlp." Advances in Neural Information Processing Systems 34 (2021): 15787-15800.

---

### Official Review · Reviewer_fcAm · 2023-10-22

**Soundness:** 2 fair
**Presentation:** 3 good
**Contribution:** 3 good
**Rating:** 5
**Confidence:** 4

**Summary:**

This work studies the quality of synthetic data generated by LLMs. The major contribution of this work is proposing a framework to evaluate LLM's ability to generate synthetic data for specific tasks, and compare behavior across different LLMs. The evaluation framework consists of five different axes: performance, complexity, conformity, diversity and faithfulness. These properties are either evaluated using accuracy-based metrics, or modified version of existing tools (e.g., distict-n, mauve, etc.). Using this framework, this work compares LLMs with different size, from different model families and with or without instruction tuning. The empirical study reveals interesting tradeoffs among the five axes, and also report general performance trends on overall performance.

**Strengths:**

1. Generating synthetic datasets is a very popular application of LLMs. This work provides a useful framework on evaluating this ability of LLMs.
2. The empirical study shows interesting tradeoff from the models, and the reported performance trends can be useful for related applications.

**Weaknesses:**

1. I like the general idea of the proposed evaluation framework, but my biggest concern about this framework is the heavy use of DistilBERT accuracies in the evaluation framework. For the faithfulness metric, the framework is evaluating the performance of DistilBERT on the generated dataset. This confounds faithfulness with the difficulty (or complexity) of the dataset. This makes some of the finding questionable. For example, is there really a tradeoff between faithfulness and diversity/complexity, or is this correlation comes from the correlation between difficulty and diversity/complexity? I wonder if the authors can provide gold evaluation results for the DistilBERT models.
2. This study only focuses on synthetic data generation for relatively simple classification tasks. It would be great if this work can include evaluation on some more complex tasks.
3. While this paper proposes four other properties addition to the performance. There is not much discussion on the relationship between these properties and the final performance. So while this study show many interesting findings, it is unclear what users should do besides checking the performance rankings.

**Questions:**

1. For the value k in the diversity metric, are you keeping the example size the same, or the token size same?
2. How do design or select prompts for the study conducted in your paper? Have you checked the sensitivity of the findings with respect to different prompts?

---

> ### Author Response · Authors · 2023-11-21
>
> We thank the reviewer for their insightful questions, which we address below. We are delighted to read that they find our work to be a useful framework and appreciate our empirical study.
>
> **Q: Is there a tradeoff between faithfulness and diversity/complexity, or does this correlation come from the correlation between difficulty and diversity/complexity?**
>
> We can reason about the potential effect of difficulty as a potential confounding variable to validate that these tradeoffs are not caused by this behavior.
>
> Regarding the tradeoff between faithfulness and diversity we note that when we increase the model's temperature to boost diversity, it starts choosing less likely words. This could make the output more difficult for a classifier to label correctly, therefore causing an increased difficulty to potentially reduce faithfulness. However, it is much more likely that the output becomes unfaithful because choosing less likely words doesn't necessarily equal creating complex or difficult content. It is more about straying from the expected or accurate output. The observation that this tradeoff has a similar slope across all vanilla models further supports this argument, since one would expect the slope to be significantly different for smaller models, which are less likely to generate correct but difficult samples.
>
> When we consider the tradeoff between faithfulness and complexity, we note that in a perfect scenario, a model that is trained on synthetic data should perform identically to one trained on a reference dataset. In such a case, the relationship between faithfulness and complexity would be straightforward: faithfulness would be the inverse of complexity. Therefore, what is really important is how actual model performance deviates from this ideal. For example, if a dataset results in a lower complexity than what this ideal relationship predicts, it implies that the data generated is simpler than expected, given its level of faithfulness. A difficult sample, being defined as a sample example that is misclassified by both classifiers, is irrelevant for this comparison since the inclusion or exclusion of this sample would not change the deviation from the ideal scenario.
>
> **Q: Can you provide gold evaluation results for the DistilBERT models?**
>
> We included the values of the reference data in Figure 2 and Table 2 and included the metrics for each dataset separately in Table 9 in Appendix E.
>
>
> **Q: Why did you not include some more complex tasks?**
>
> We specifically selected our tasks to evaluate the data generation capabilities of language models on common data domains. This allowed us to investigate the difference between the modeling capabilities of vanilla models and instruction-tuned models. Dataset generation for more complex tasks would significantly complicate the evaluation, since they typically entail advanced prompting techniques, and instruction-following capabilities would become essential for generating faithful samples. We also note that we use our metrics as a proxy to measure the data quality in general and not necessarily just for the downstream task of classification.
>
> **Q: What are the recommendations for practitioners to use LLMs for training data generation?**
>
> Please see Q1 of our main reply.
>
>
> **Q: For the value $k$ in the diversity metric, are you keeping the example size the same, or the token size the same?**
>
> We are keeping the token size the same. Since samples generated by different models might on average differ in size, keeping the sample size the same is not enough to compensate for the size dependency of the used metric.

---

> > ### Author Response · Authors · 2023-11-21
> >
> > **Q: How do you design or select prompts for the study conducted in your paper? Have you checked the sensitivity of the findings with respect to different prompts?**
> >
> > Our (very simple) prompts were selected after ensuring that each model could follow the instructions and generate reasonable completions by manual inspection. This is done by a simple sanity check. Specifically, we generated a number of samples (typically around five) using the prompt and if all of these remained "on topic", we then used that prompt.
> >
> > We note that the goal of prompts is different for dataset generation than in other areas of machine learning, such as Q&A and reasoning. While in these areas, there is only a single correct answer and the prompt needs to be optimized for that, in data generation the purpose of the prompt is to guide the language model towards the correct distribution. While different prompts could result in different distributions, we do not believe this influences any of our results. This claim is validated by the fact that our conclusions hold over multiple model families and models. If our conclusions were sensitive to the specific prompt used, one would expect that the use of different models, which could behave differently under the same prompt, would result in different conclusions. Furthermore, we note that the cost and time associated with generating these datasets is high, with dataset generation for a single model taking several days on an H100 GPU, it is thus not feasible to perform extensive multi-model prompt tuning.
> >
> > We hope to have been able to address all the reviewers’ concerns, are happy to answer any follow-up questions they might have, and are looking forward to their reply.

---

### Official Review · Reviewer_wzFz · 2023-10-30

**Soundness:** 4 excellent
**Presentation:** 3 good
**Contribution:** 3 good
**Rating:** 6
**Confidence:** 4

**Summary:**

This paper studies the text generation capabilities of various large language models, proprietary and open, instruction-tuned and vanilla, by evaluating synthetic datasets generated from them. The datasets are evaluated in terms of
1) diversity in vocabulary
2) complexity, or difficulty in modeling them given by the performance of a model trained and evaluated in-distribution.

By comparing the generated datasets to existing (reference) datasets in similar tasks and domains, they are also evaluated in terms of
3) faithfulness, given by the performance of models trained on the reference datasets and evaluated on the generated ones
4) conformity, given by a measure of distributional similarity between the reference and generated datasets
5) performance, given by the performance of models trained on the generated datasets and evaluated on the reference datasets

Based on this evaluation framework, the paper discusses the tradeoffs between these aspects of generation quality, how they change across model families, and how instruction tuning affects these tradeoffs.

**Strengths:**

The evaluation framework is sensible and analyzing the capabilities of language models in terms of the tradeoffs between various aspects of generation quality is quite informative. The results of studying the effect of model size, the impact of instruction tuning, and that of the level of instruction tuning can potentially inform how to finetune future versions of language models.

**Weaknesses:**

This study has some missing details, several limitations, and potential confounders not accounted for in the experiments.

Missing details

MD1:The evaluation is done over four classification datasets, but the actual details of the tasks are missing in Section 4. Particularly for AGNews and ELI5, it is unclear what is being classified After reading the Appendix, the AGNews task seems to be some news genre classification, and the ELI5 task seems to be subreddit classification (maybe it should just be called "subreddit classification"?) This issue can easily be fixed by including explicit details in Section 4.

MD2: The motivation behind the chosen evaluation metrics is somewhat unclear. Particularly, faithfulness, conformity, and performance seem to be measuring the difference between the generated and reference data distributions. Why do we need these three variants? Relatedly, one would expect these metrics to correlate highly with each other. Analyzing this further would be helpful.

Limitations and potential confounders

L1: It is unclear how noise in the datasets (due to inaccurate labels) affects the trends seen in tradeoffs. For example, is the increase in diversity beyond the the conformity threshold in Fig 2 simply be due to noise? Having humans classify (subsets of) the generated datasets, and introducing the accuracy of the synthetic datasets as an additional metric could make this clearer.

L2: The biases in the reference datasets could also be affecting conformity, faithfulness and performance. It might help to include multiple reference datasets per domain-task combination to evaluate whether the trends hold across them.

L3: It is possible that the models used for generating datasets have seen the reference datasets either during pretraining or instruction-tuning. This would inflate the quality measures according to conformity, faithfulness, and performance. This issue cannot be dealt with directly, but it would help to check the zero-shot performance of the large language models on the reference datasets, and take it int account while inferring the tradeoffs.

**Questions:**

- It would be helpful to put the reported diversity and complexity values in context. What are these values for the reference datsets?
- Can you elaborate on the motivation behind the three metrics comparing generated and reference datasets (see MD2)?

---

> ### Author Response · Authors · 2023-11-21
>
> We thank the reviewer for their insightful questions, which we address below. We are delighted to read that they find our work to be a useful framework and appreciate our empirical study.
>
> **Q: Can you include specific details of the classification tasks in Section 4?**
>
> We have now specified what purpose each classification task has in Section 4.
>
> **Q: Why do we need complexity, faithfulness and performance?**
>
> These three characteristics are all measured as an accuracy of a classifier trained and evaluated on different data. Intuitively, these metrics measure three separate and interesting characteristics of the datasets: faithfulness for how correct the generated samples are, complexity for how difficult it is to fit a model on the generated dataset (and therefore how complex it is) and performance for the final performance of the dataset on the downstream task. Furthermore, as discussed in section 4.1, the interaction between these metrics is quite interesting and shows interesting tradeoffs that exist between them.
>
> First, faithfulness and complexity are very highly correlated as noted in the third tradeoff we discuss in Section 4.1. The high correlation between faithfulness and complexity is a consequence of their interaction. In an ideal scenario, the model trained on the reference dataset would be the exact same as the one trained on the synthetic dataset. Thus, faithfulness would be exactly equal to $1 - \text{complexity}$. Therefore, the interesting aspect of this behavior relates less to the tradeoff itself, but more to the deviations from this ideal scenario, as discussed in the paper. A specific dataset can have a lower complexity than the one expected as the ideal outcome, indicating that the resulting data is quite simple for the amount of faithful samples it contains.
>
> Second, the dependence of the downstream performance on faithfulness and complexity is solely through their sum. This is caused by the above discussed tradeoff and now more thoroughly discussed in Section 4.1.
>
> **Q: How does the noise in the generated datasets affect the results? Can it have an influence on the diversity/conformity tradeoff?**
>
> The key problem of synthetic datasets is their inherent noise, which is crucial for understanding and analyzing them. We have designed specific characteristics in our approach to identify different types of noise in these datasets. For example, faithfulness is a measure of noise resulting from incorrect labeling, whereas conformity measures noise caused by shifts in style and tone.
>
> Our findings, particularly the tradeoff between diversity and conformity, show the interplay of different noise types within the dataset. When diversity is low, the difference in style between the synthetic and reference datasets is a consequence of a too narrow generation of the data distribution and results in low conformity. On the other hand, high diversity often leads to the inclusion of irrelevant examples, which contributes to the style shift and therefore results in a low conformity. This interaction results in a quadratic relationship between diversity and conformity in the generation process.
>
> **Q: Why did you not include several reference datasets for each data domain to study the effects of the biases on conformity, faithfulness and performance?**
>
> While biases in the reference datasets could affect these three metrics, we do not believe this has significant influences on the conclusions drawn in the paper. We chose four common data domains with a clearly defined style and purpose. The chosen datasets are widely known and form a representative sample of the specific data domain and while it is possible that several nuances are not captured as well as they could, it is unlikely that these affect the classification accuracy in a significant way.
>
> Furthermore,  the conclusions drawn in the paper hold over all data domains and for all models, indicating that even if there were biases, they are consistent throughout the generation process and do not influence our characteristics. The conclusions drawn from the data have an intuitive explanation, e.g., the better conformity and diversity of vanilla models with respect to their instruction-tuned counterparts. This also indicates that biases do not play a significant role in the generation process.
>
> **Q: What would the influence of models having seen these datasets during training be?**
>
> It is likely that parts of these datasets were included in the training data of the models. This is intentional, as we want to measure how effectively the models have learned the distribution associated with these common data domains. This approach enables us to evaluate and compare the performance of both standard (vanilla) and instruction-tuned models. Additionally, it helps us to identify and analyze the changes in style that result from instruction tuning and RLHF.

---

> > ### Author Response · Authors · 2023-11-21
> >
> > **Q: Can you provide the values for the reported diversity and complexity values for the reference datasets?**
> >
> > We added the values of the reference data in Figure 2 and Table 2 and included the metrics for each dataset separately in Table 9 in Appendix E.
> >
> > We hope to have been able to address all the reviewers’ concerns, are happy to answer any follow-up questions they might have, and are looking forward to their reply.

---

### Official Review · Reviewer_PKWU · 2023-10-30

**Soundness:** 3 good
**Presentation:** 4 excellent
**Contribution:** 3 good
**Rating:** 8
**Confidence:** 4

**Summary:**

This work studies the attributes of dataset generation, which has recently been explored as a way to train task networks without needing a natural, human-generated dataset. Particularly, this work studies 4 domains/tasks that dataset generation can be applied to (e.g. SST-2), and studies the trade-offs between different attributes: faithfulness, diversity, conformity, complexity, and performance, all of which the authors measure automatically. The authors find significant differences between different model types, especially finding that instruction-tuned models differ from classical LMs. Neither paradigm seems to completely dominate.

**Strengths:**

- Overall, this type of contribution is sorely needed in dataset generation, which is still not a well-understood field
- The attributes to study are diverse and relevant
- Very interesting and informative conclusions drawn about the tradeoffs, e.g. the loss of diversity in generated datasets when using instruction-tuned models
- paper is well presented and quite clear

**Weaknesses:**

- I have concerns wrt the measurement of some of the aspects:
   - faithfulness is measured as the accuracy on the synthetic set with a model trained on the reference (human) set. While being unfaithful is one reason this value may be low, it is not the only one. It is easy to imagine a *faithful* dataset on which this classifier will perform poorly, due to issues like style shift or poor generalization of the classifier. To be more concise: staking faithfulness on the accuracy of a classifier ignores the fact that this may be an issue of the classifier rather than the dataset that is being evaluated.
   - similar issue with complexity, which is measured as inverse accuracy on a held out chunk of the synthetic set. While I agree that lower complexity will indeed raise this accuracy, high complexity is not the only reason this accuracy may decrease.
- Overall, I would suggest renaming these metrics. They likely correlate with the values they are described as, but it is overly presumptuous to label them this way as there are many other factors. More direct names (e.g. complexity -> self-accuracy or something like this) might be more accurate, leaving discussion of factors affecting these values (like complexity) to the discussion
- Tradeoffs (Figure 2) are only shown in terms of temperature, which may be a confounding factor. It would be good to show other curves, e.g. for values of top-p, because it is not clear if these tradeoffs may have to do specifically with the specific warping effect that temperature has on sampling distributions. Alternatively, being more precise in the paper text, that these are tradeoffs over temperature as the variable, rather than general tradeoffs.

**Questions:**

Have you tried variables besides temperature to test the tradeoffs?

---

> ### Author Response · Authors · 2023-11-21
>
> We thank the reviewer for their insightful questions, which we address below. We are delighted to read that they find our work to be sorely needed, well-written and our results to be informative and interesting.
>
> **Q: What is the influence of style shift or poor generalization of the classifier on the faithfulness and complexity metrics?**
>
> They are an essential part of the proposed metrics for these characteristics. Regarding faithfulness, we specifically chose this name rather than correctness, to indicate that the classifier accuracy is not only dependent on the correctness of a label, but also whether or not the associated sample is faithful or in line with reference data samples. A lower faithfulness can therefore be caused by either incorrect labels or samples that are so far from reference samples that they cannot be classified correctly. While poor generalization could impact the measured faithfulness values, this issue would uniformly affect all models evaluated in our study, thereby not changing the outcomes or conclusions.
>
> Regarding complexity, we note that we introduce it as the complexity associated with the entire dataset rather than the complexity associated with specific samples. Simple datasets have a higher generalization error which is a problem due to the high variety in real-world samples. While complex or difficult samples is one reason for an increased complexity of a dataset, it could for example also be caused by a wider variety of samples.
>
> We have now further emphasized these effects in Section 3 of the paper, where they are first introduced.
>
>
> **Q: Can you show the tradeoffs against other variables besides temperature?**
>
> Great question!
> While our experiments include many variables (model size, model family, temperature, etc.) that are interesting, we use temperature for the plots as (i) we find it to be the main driver of the trade-offs, and (ii) it is a continuous variable and therefore lends itself to plotting.
>
> The only other continuous variable is model size via parameter count, which we include in the new Appendix D where we find that the observed tradeoffs are the same when the dependent variable is model size instead of temperature. This shows that the observed tradeoffs are not specifically due to a warping effect on the output distribution introduced by changing the temperature.
>
> Further, we find that categorical variables reveal key-tradeoffs: Comparing vanilla and instruction-tuned models on the (existing temperature-base) plots, we find that instruction-tuned models have lower diversity and higher faithfulness for the same temperature. However, the plots show that when compensating for temperatures, instruction-tuned models generally perform better on this tradeoff as they are more at the top right of the plot. Similarly, we find that Falcon-7b performs worse on both the diversity vs faithfulness and diversity vs conformity tradeoffs, indicating that it is not such a powerful model.
>
> Additionally, for all open-source and GPT3-175B models and for each data domain, we now also generated datasets using nucleus sampling with a parameter of 0.9 and added them to all relevant figures in the paper, i.e., Figure 2, Figure 5 and Figure 6. As can be seen there, this extra point follows all the curves formed by the temperature and therefore supports the conclusions made in the paper.
>
> We hope to have been able to address all the reviewers’ concerns, are happy to answer any follow-up questions they might have, and are looking forward to their reply.

---

> > ### Comment · Reviewer_PKWU · 2023-11-23
> >
> > Thank you for a comprehensive response, this answers my questions.

---

### Author Response · Authors · 2023-11-21

We thank the reviewers for their feedback and comments. In particular we are delighted that they found that the studied task is interesting (Reviewer Afrd), is sorely needed in dataset generation (Reviewer PKWU) and that our findings are interesting (Reviewer fcAm) and informative (Reviewer PKWU, Reviewer wzFz).

Here we briefly outline the changes made to the manuscript, mostly clarifications, and a recurring point in the reviews. We address other questions in individual replies.

**Changes to the Manuscript**

- In Appendix D, we analyze the tradeoffs between complexity, diversity, faithfulness and conformity over model size (rather than temperature in Section 4.1).
- Added further clarification in Section 3 for the purpose of introducing faithfulness and complexity and what aspects of the synthetic dataset they are influenced by.
- Discussed the third observed tradeoff in Section 4.1 between faithfulness and complexity more accurately and described the exact nature of this tradeoff.
- Included the value of the characteristics on the reference dataset in Figure 2 in Section 4.1, Table 2 in Section 4.2 and Table 9 in Appendix E.
- Made minor edits to fix typos and enhance grammar.

**Q1: Based on your work, what are the recommendations for practitioners to use LLMs for training data generation?**

We encourage practitioners to pay attention to all characteristics of a specific dataset and to look further than just downstream performance and diversity. Recall that all of our characteristics measure concrete effects and biases introduced to the generated datasets, indicating that a wider and more detailed view on dataset generation can be beneficial to many downstream tasks.

Concretely, we recommend practitioners to consider the specific use cases when selecting LLMs for training data generation. While ChatGPT is a popular choice, it may not always be the most suitable. For instance, if dataset conformity is a priority, choosing vanilla models could yield better results. Each model introduces its unique biases and strengths to the generated data, e.g., by generating faithful but non-conform samples for instruction-tuned models. Therefore using a combination of different LLMs could increase performance and mitigate problems or biases related to a specific model.

Another important aspect for dataset generation is the sampling temperature for the generated dataset. Changing this temperature can significantly alter results, and we therefore encourage practitioners to test different sampling temperatures for optimal performance.

Finally, we would encourage the inclusion of examples in the prompt. We found that especially the conformity and diversity of instruction-tuned models increases dramatically when introducing few-shot samples and found it also improved the performance on the downstream task.

---

### Meta-Review · Area_Chair_w8dC · 2023-12-14

**Metareview:**

This paper discusses multiple automatic metrics to evaluate synthetic data generated by LLMs, considering different models and training regimes, namely: faithfulness, diversity, conformity, complexity, and performance. Findings reveal differences in instruction-tuned models for data generation vs other model families, where they find a loss of diversity.

Strengths: This paper presents valuable work towards evaluating generations of language models, and reveals important insights into what might be determining the quality of generated data under different training regimes, especially instruction tuning which has led to many generated datasets.

Weaknesses: Reviewers pointed out some issues with the selection and definition of the metrics, and lack of empirical evidence to conclusively infer some of the claims.

**Justification For Why Not Higher Score:**

Please see weaknesses; there were some issues with the definitions of some metrics, and some experimental settings which were not fully explored.

**Justification For Why Not Lower Score:**

n/a

---

### Decision · Program_Chairs · 2024-01-16

Reject